# A Bait-and-Hook Hydrogel for Net Tumor Cells to Enhance Chemotherapy and Mitigate Metastatic Dissemination

**DOI:** 10.3390/pharmaceutics16121516

**Published:** 2024-11-25

**Authors:** Cailian Chen, Jinying Liu, Hongbo Zhang, Hongrui Zhang, Yanhui Liang, Qilian Ye, Wei Shen, Haibin Luo, Ling Guo

**Affiliations:** 1Key Laboratory of Tropical Biological Resources of Ministry of Education, School of Pharmaceutical Sciences, Hainan University, Haikou 570228, China; 22220860020015@hainanu.edu.cn (C.C.); 22211007000008@hainanu.edu.cn (J.L.); 23110710000007@hainanu.edu.cn (H.Z.); 22211007000029@hainanu.edu.cn (Q.Y.); 22211007000026@hainanu.edu.cn (W.S.); 2Department of Neurosurgery, The Second Affiliated Hospital of Nanchang University, Nanchang 330006, China; hongbozhang99@smu.edu.cn; 3Institute of Drug Testing, Hainan Academy of Inspection and Testing, Haikou 570311, China; 23211007000014@hainanu.edu.cn

**Keywords:** bait-and-hook drug delivery, anti-metastatic hydrogel, CXCL12, tumor capture, doxorubicin liposomes

## Abstract

**Background**: Lung cancer is an aggressive disease with rapid progression and a high rate of metastasis, leading to a significantly poor prognosis for many patients. While chemotherapy continues to serve as a cornerstone treatment for a large proportion of lung cancer patients, expanding preclinical and clinical evidence indicates that chemotherapy may promote tumor metastasis and cause side effects. **Methods**: We develop an injectable bait-and-hook hydrogel (BH-gel) for targeted tumor cell eradication, which embedded doxorubicin liposomes as cytotoxic agents and CXCL12 as a chemoattractant to capture and kill tumor cells. The hydrogel backbone was formed through covalent cross-linking between PVA and borax. In vitro, we investigated tumor recruitment and the antitumor effects in A549 cells. In vivo, we explored the anti-metastatic and antitumor activities against lung cancer. **Results**: BH-gel retained CXCL12 within its three-dimensional porous architecture for gradual release, effectively recruiting tumor cells. In contrast, blank hydrogel failed to achieve this. After encapsulation in BH-gel, the therapeutic efficacy of doxorubicin liposomes for tumor eradication was markedly improved, significantly reducing metastatic tumor presence to near-undetectable levels, while also resulting in notable reductions in cardiotoxicity and hepatotoxicity. Notably, BH-gel adhered well to tissues and exhibited exceptional electrical conductivity, which may be further developed into a real-time tumor monitoring system, facilitating timely therapeutic adjustments. **Conclusions**: BH-gel utilizes CXCL12 as a bait to recruit and entrap tumor cells in a three-dimensional porous matrix and subsequently kill them with embedded doxorubicin liposomes, thereby tackling the issue of metastatic spread. This bait-and-hook strategy has significant implications for the field of anti-metastasis medicine and shows considerable potential for clinical application.

## 1. Introduction

Lung cancer ranks as the leading cause of cancer-related deaths worldwide, with the highest incidence and mortality rates among all types of cancers [1,2]. Despite significant advancements in early detection and treatment, individuals with advanced lung cancer still face a grim prognosis, largely due to the disease’s tendency for rapid metastasis [3]. Cancer metastasis to distant sites is strongly associated with poor patient prognosis and is the leading cause of cancer-related deaths, with around 90% of cancer deaths resulting from metastatic disease [4]. Therefore, the significance of targeting tumor metastasis in therapy cannot be overemphasized [5].

As the most widely accepted treatment for lung cancer, chemotherapy induces cell death through various mechanisms such as antimetabolites, alkylating agents, and mitotic spindle inhibitors [6]. Numerous drug delivery systems have been developed to enhance the therapeutic efficacy of chemotherapy, including liposomes, nanoparticles, and hydrogels [7,8]. In clinical drug development, the primary focus is generally on tumor shrinkage, as assessed by the radiological Response Evaluation Criteria for Solid Tumors (RECIST), while the capacity to inhibit metastasis is frequently disregarded [9]. It is only after clinically significant tumor responses or improvements in patient survival are observed in metastatic cases that a drug progresses to adjuvant trials aimed at delaying the emergence of overt metastasis. There remains a shortage of both preclinical research and clinical development for therapies that directly target the metastatic process [10].

In addition to lacking anti-metastatic properties, traditional chemotherapeutic drugs can even trigger and promote metastasis, which significantly diminishes their clinical efficacy [11]. The underwhelming clinical results can be attributed in part to the inability of cytotoxic agents to capture the complexity of a disease characterized by multiple subtypes. Pronounced intratumoral heterogeneity further complicates treatment, leading to inconsistent responses to targeted therapies [10]. Effective metastasis treatment requires inhibiting the core mechanisms of metastasis and designing preclinical and clinical strategies that focus on more than just halting cancer cell proliferation [12]. There are two complementary anti-metastasis strategies, the prevention of cancer cell proliferation and the suppression of tumor escape [13]. Migrastatic strategies offer a new form of treatment for metastasis, providing a distinctive therapeutic approach aimed at preventing various types of cancer cell migration and invasion [14]. Since the migration machinery has been shown to facilitate metastatic dissemination, a successful migrastatic therapy administered alongside standard treatment could have a synergistic effect, which potentially reduces the risk of metastasis. The most promising targets for migrastatic therapy are likely those essential for cellular motility [15]. CXCL12, secreted by stromal cells, is known to attract cancer cells by stimulating the CXCR4 receptor, which is upregulated in lung tumor cells [16]. The CXCR4-CXCL12 chemokine signaling axis is instrumental in promoting the formation of distant metastases. Elevated release of CXCL12 in prevalent metastatic sites can draw cancer cells and enhance their migration to these locations [17]. Therefore, leveraging CXCR4 expression on metastatic cells, we propose that the development of an injectable gel containing CXCL12 could regulate tumor cell migration direction, preventing their escape to distant sites. This approach may help avoid drug-induced metastasis and demonstrate a synergistic effect with cytotoxic agents.

Here, we present an injectable bait-and-hook hydrogel (BH-gel) created via covalent cross-linking between polyvinyl alcohol (PVA) and borax. This hydrogel serves as a carrier for the local delivery of doxorubicin liposomes (Dox Lipo) to improve lung cancer treatment in vivo, employing CXCL12 as a chemoattractant to effectively capture and engage tumor cells. Rheological data demonstrated that the BH-gel exhibits injectable shear-thinning properties and exceptional electrical conductivity. To evaluate its therapeutic potential, we examined key properties of the BH-gel, including tumor recruitment and Dox-induced invasion, which are essential for effectively suppressing metastatic lung cancer. The antitumor efficacy of the BH-gel was assessed in a mouse model of metastatic lung cancer. These findings provide insights into an innovative bait-and-hook hydrogel aimed at mitigating metastatic dissemination.

## 2. Materials and Methods

### 2.1. Materials

Doxorubicin hydrochloride (Dox HCl) was purchased from MedChem Express (Monmouth Junction, NJ, USA) and desalted to its protonated form (Dox). Lecithin (99.8%) and cholesterol (99%) were sourced from Aladdin Reagent Co., Ltd. (Shanghai, China). Polyvinyl alcohol (PVA, 1799) and borax were obtained from Macklin Reagent Co., Ltd. (Shanghai, China). DiI and Hoechst 33342 were acquired from Beijing Biyuntian Biotechnology Co., Ltd. (Beijing, China). DMEM, fetal bovine serum (FBS), Trypsin-EDTA, and penicillin/streptomycin came from Gibco (Grand Island, NY, USA). Dimethyl sulfoxide (DMSO) and 3-(4,5-dimethylthiazol-2-yl)-2,5-diphenyl tetrazolium bromide (MTT) were sourced from Sigma-Aldrich (St. Louis, MO, USA). Leagene Biotechnology (Beijing, China) supplied 2.5% glutaraldehyde and 4% paraformaldehyde solutions. The Annexin V-FITC apoptosis detection kit was obtained from Bestbio Biology (Shanghai, China).

### 2.2. Experimental Cell Lines and Animals

The human non-small-cell lung cancer cell line A549 and a stable A549 cell line expressing green fluorescent protein (GFP) was obtained from the American Type Culture Collection. All cells were cultured in a humidified incubator (Thermo Fisher Scientific, Waltham, MA, USA) at 37 °C with 5% CO_2_. Four-week-old female BALB/c nude mice and a 10-week-old male New Zealand white rabbit were acquired from the Laboratory Animal Center at Hainan University (Haikou, China). BALB/c nude mice housed under specific pathogen-free (SPF) conditions, with continuous access to standardized food and water. All animal experiments were conducted in strict accordance with the Guiding Principles for the Use of Laboratory Animals, as approved by the Institutional Animal Care and Use Committee at Hainan University.

### 2.3. Preparation of Dox Lipo

Dox was prepared by desalting Dox HCl into its protonated form. In brief, 150 mg of lecithin, 50 mg of cholesterol, and 5.4 mg of Dox were dissolved in 15 mL of dichloromethane and stirred at 60 °C for 30 min until fully dissolved. The mixture was then transferred to a round-bottom flask, and the dichloromethane was evaporated under vacuum using a rotary evaporator at 45 °C until a film formed. Once a uniform film was created, 6 mL of preheated PBS (pH 7.4) was added. The system was hydrated in a 60 °C water bath for 1 h, ensuring complete hydration of the lipid membrane on the bottom and walls of the flask. An ultrasonic cell crusher (VCX800, Sonics & Materials, Inc., Newtown, CT, USA) was then used to homogenize the mixture (7 min, 30% power, on 1 s, off 1 s), to form uniformly sized liposomes. Finally, the Dox-loaded liposomes were filtered through 0.8 µm, 0.45 µm, and 0.22 µm Millipore syringe filters, respectively, and stored at 4 °C for further use.

### 2.4. Preparation of BH-Gel

A BH-gel containing 1.3% PVA and 0.026% borax was synthesized using a one-pot reaction method. First, PVA was dissolved in double-distilled water to a total volume of 9.0 mL. The solution was heated in an oil bath at 90 °C with magnetic stirring for 60 min until fully dissolved. After cooling to room temperature, 4.5 mL of a borax solution containing Dox Lipo and CXCL12 (6 µg/mL) was added to the PVA solution. The mixture was stirred rapidly for 5 s to promote cross-linking and form the BH-gel.

### 2.5. Molecular Docking Analysis

Molecular docking analysis was performed to study the interaction between phospholipid (PC) and Dox. Before docking, the molecular structures of PC and Dox were drawn with King Draw software (www.kingdraw.com, accessed on 10 March 2023). The molecules were preprocessed using AutoDock Tools 1.5.6 (http://autodock.scripps.edu/resources, accessed on 10 March 2023) by adding hydrogens and charges. The binding site was defined as a spherical region with a 20 Å radius, sufficient to encompass all potential binding regions. Docking was conducted using AutoDock Vina (version 1.1.2), and the result with the lowest binding energy was selected for further analysis. Interactions were visualized with PyMOL (http://www.pymol.org, accessed on 10 March 2023).

### 2.6. Analysis of the Circular Dichroism (CD) Spectra of Dox and PC

PC solutions with different concentrations of Dox were prepared in aqueous solution containing 67% ethanol. The mixture was placed in a micro cuvette with a 1 cm path length. CD spectra were recorded in the wavelength range of 200–250 nm, with a 1 nm step size and a measurement time of 0.5 s per point. The data were processed as previously described for the dimer.

### 2.7. Particle Characterizations of Dox Lipo

The hydrodynamic diameter, polydispersity index (PDI), and zeta potential of Dox Lipo were measured using dynamic light scattering (Malvern Panalytical ZSU3100, London, UK). The stability of Dox Lipo was assessed over 14 days by monitoring changes in particle size and PDI. The morphology of Dox Lipo was visualized with a field emission transmission electron microscope (Talos F200X G2, Thermo Fisher Scientific, Brno, The Czech Republic) using a negative staining technique with 1% phosphotungstic acid.

### 2.8. Leakage Profile of Dox in Dox Lipo

Leakage profile of Dox Lipo was determined via ultrafiltration centrifugation. The absorbance of varying concentrations of free Dox in DMSO was measured at 481 nm in a UV spectrophotometer (UV-2600, Shimadzu, Kyoto, Japan) to generate a standard curve. Then, a specific amount of Dox Lipo was dispersed in DMEM containing 10% FBS and incubated at 37 °C in a humidified incubator. At predetermined intervals, the release medium was centrifuged at 300× *g* for 5 min. The precipitation was analyzed for free Dox, which was resuspended in 300 µL of DMSO and vortexed for 1 min for quantification. Dox concentration was measured via absorbance at 481 nm using a UV spectrophotometer. The leakage rate was calculated as the percentage of Dox released from Dox Lipo over time.

### 2.9. Fourier Transform Infrared (FT-IR) Absorption Spectroscopy

In order to study the gelation mechanism of BH-gel, a certain amount of PVA, borax, and blank gel were prepared and lyophilized, mixed with potassium bromide, and pressed with a tablet press. The spectra of the samples were acquired using FT-IR (Nicolet 5700 FT-IR Spectrometer, Thermo Fisher Scientific, Waltham, MA, USA) via the pressed slice method, covering a wavelength range of 4000 to 400 cm^−1^.

### 2.10. Gelation Time

The gelation time of the BH-gel was determined using the vial inversion method. A mixture of PVA and borax solutions was placed in a vial and then tilted and inverted. The gelation time was recorded when no flow was observed in the inverted vial.

### 2.11. Morphological Structure Observation of BH-Gel

Cross-sections of the lyophilized BH-gel were mounted onto a metal pedestal and coated with a gold layer under vacuum. The morphological structures were observed using a scanning electron microscope (SEM, EVO MA10, ZEISS, Baden-Württemberg, Germany).

### 2.12. Rheological Properties

The rheological properties of BH-gel were analyzed using a rheometer (Kinexus Lab+, Malvern, UK). Frequency sweeps (0.1–10 Hz at 1% strain) were performed to evaluate changes in the storage modulus (G′) and loss modulus (G″). Adynamic rheological test sweeping was conducted from 0.1% to 100% strain at 25 °C. Continuous step-strain measurements were also conducted to assess the self-healing capability of the BH-gel. The BH-gel was alternately subjected to 100% strain to disrupt the hydrogel network, followed by 10% strain to enable restoration. The viscosity of BH-gel was measured across shear rates from 0.1 to 100 s^−1^.

### 2.13. Injectability and Shape Adaptability

The injectability of the BH-gel was assessed using a syringe fitted with a needle. To demonstrate its flow properties, Rhodamine B-dyed hydrogel was extruded to form the letters “HNU” on cardboard at room temperature. The shape adaptability of the BH-gel was tested by injecting it into a circular mold. Furthermore, the flexibility of the BH-gel was evaluated by being stretchable into various shapes and adhering to porcine skin.

### 2.14. Migrating Kinetics of Tumor Cells

The in vitro migration kinetics of tumor cells were evaluated using CIM-16-well plates with an xCELLigence RTCA-DP instrument (Roche Diagnostics, West Sussex, UK). BH-gel without doxorubicin was loaded into the lower wells of the CIM-16 plate. The upper chamber was attached, and the upper wells were filled with 30 μL of prewarmed medium, followed by 30 min of pre-equilibration. Next, 100 μL of A549 cell suspension (2 × 10^4^ cells) was added to each top well. The plate was transferred to the RTCA-DP machine, where data were collected every 5 min over 420 sweeps (35 h in total). The electrical impedance signal, reflecting the cell index, was captured in a representative trace. An increase in impedance corresponded to more A549 cells migrating and adhering to the bottom of the lower chamber. Migrated cells were stained with Calcein-AM, and live-cell images were obtained using the cell imaging system (EVOS, M7000, Thermo Fisher Scientific, Waltham, MA, USA).

### 2.15. BH-Gel Biosafety Assessment on Rabbit Ears

In the BH-gel biosafety experiment on rabbit ears, a 10-week-old male New Zealand white rabbit was chosen as the subject due to its well-characterized anatomy and sensitivity, which allows for an effective evaluation of dermatological responses. Prior to the application, the inner ear of the rabbit was photographed and documented as baseline images. Then, 1 mL of BH-gel without doxorubicin was applied to the rabbit’s inner ear and the rabbit ear was closely monitored for any adverse reactions.

### 2.16. In Vivo Tumor Recruitment Experiment

BALB/c nude mice were subcutaneously injected in the flank with 1.0 × 10^6^ A549 cells on day 0. Then, the hydrogel was injected into the area adjacent to the tumor using a needle (BD Biosciences, Franklin Lakes, NJ, USA). After 2 days, the gels were removed for observation and photography. To better evaluate the recruitment ability of BH-gel, blank liposomes were incorporated into the BH-gel.

### 2.17. BH-Gel Degradation Behavior and Dox Lipo Release In Vitro

The BH-gel was weighed (W_0_) and immersed in 10 mL of PBS (pH 7.4). The samples were gently shaken in an incubator shaker at 37 °C and 100 rpm for 48 h. At various time points, photographs were taken, and the remaining weight of the gels (W_t_) was measured. The remaining weight ratio was calculated using the following formula:$$\mathrm{Weight}\;\mathrm{ratio}\;(\%)=W_{t}÷W_{0}\times 100\%$$

The BH-gel (containing 0.3 mg of Dox) was immersed in 10 mL of PBS (pH 7.4) and gently shaken in an incubator shaker at 37 °C and 100 rpm for 48 h. At various time points, 1 mL of leachate was collected and replaced with 1 mL of fresh PBS. The leaching solution was centrifuged and demulsified with DMSO. The Dox Lipo concentration in the leachate was measured via absorbance at 481 nm using a UV spectrophotometer (UV-2600, Shimadzu, Kyoto, Japan). The release rate of Dox Lipo from the gel was then calculated.

### 2.18. Cellular Uptake Test

The cellular uptake of Dox Lipo by A549 cells was assessed using flow cytometry and confocal laser scanning microscopy. Briefly, A549 cells were seeded in 12-well plates and incubated at 37 °C until reaching 60–80% confluence. The culture medium was then replaced with fresh medium containing Dox Lipo at a final concentration of 32 μM, and the cells were incubated for 4 h. After incubation, the medium was discarded, and the A549 cells were harvested. The cells were washed three times with cold PBS and resuspended in PBS for flow cytometry analysis (CytoFLEX, Beckman Coulter, Miami, FL, USA). Similarly, A549 cells were seeded in glass-bottom dishes and incubated at 37 °C until 60–80% confluence was reached. The culture medium was replaced with fresh medium containing Dox Lipo at a final concentration of 32 μM, and the cells were incubated for 4 h. After discarding the medium, Hoechst 33342 was added for a 10 min incubation. The cells were then washed three times with cold PBS and imaged using confocal laser scanning microscopy (FV3000, OLYMPUS, Tokyo, Japan).

### 2.19. Cytotoxicity Assay

A549 cells were seeded at a density of 5000 cells/well in 96-well plates (Costar, Corning, NY, USA) and incubated for 24 h. When cells reached approximately 80% confluence, the medium was replaced with fresh medium containing Dox Lipo. After a further 24 h of incubation, 10 μL of MTT solution (5 mg/mL in PBS) was added to 100 μL of medium per well. The plate was then incubated for 4 h at 37 °C until purple formazan crystals were clearly visible. Then, the culture medium was removed, and 150 μL of DMSO was added to each well to dissolve. Absorbance was measured at 570 nm using a microplate spectrophotometer (Epoch, Bio-Tek, Winooski, VT, USA).

### 2.20. Detection of Apoptosis

A549 cells were seeded into a 24-well plate at a density of 3 × 10^4^ cells per well. After 24 h of treatment with different conditions (Dox, Dox Lipo), the cells were processed according to the Annexin V-FITC Apoptosis Detection Kit protocol (Solarbio Inc., Beijing, China). The treated cells were subsequently analyzed via flow cytometry.

### 2.21. Tumor Model In Vivo

A tumor model was established in BALB/c nude mice via intraperitoneal injection. Briefly, 5 × 10^6^ A549-GFP cells were suspended in 25 μL of culture medium and Matrigel (BD Biosciences, Franklin Lakes, NJ, USA) and then injected into the intraperitoneal space near the intestine. After tumor cell injection, the mice were placed in the right lateral decubitus position and monitored for 30 min until full recovery. Tumor-bearing mice received intraperitoneal injections of BH-gel (2 mg/kg, every 3 days), with body weight recorded. In vivo fluorescence imaging was conducted on days 1, 7, 14, 21, and 28 to track tumor migration using the Night OWLIILB 983 system (Berthold Technologies, Baden-Württemberg, Germany) with a 5 s exposure time. Photographic and fluorescent images were captured separately and then overlaid. After 4 weeks, the mice were euthanized, and major organs (heart, lungs, liver, spleen, and kidneys) along with tumor tissue were collected and weighed. The intestine was examined for metastatic nodules. Hearts from each group were further embedded in paraffin for hematoxylin and eosin (H&E) staining.

### 2.22. Serum Biochemical Analysis

Blood was drawn from the mice via the orbital venous plexus, with approximately 1 mL collected from each mouse. Blood samples were allowed to clot for 15–30 min at room temperature and then centrifuged at 3000 rpm for 20 min to separate the serum. Serum levels of alanine aminotransferase (ALT), aspartate aminotransferase (AST), alkaline phosphatase (ALP), and total bilirubin (TBIL) were measured using an automatic biochemical analyzer (HITACHI, Tokyo, Japan).

### 2.23. Statistical Analysis

All statistical analyses were performed using Prism 9.5.1 software (GraphPad Software Inc., La Jolla, CA, USA), with unpaired Student’s *t*-tests and one-way or two-way ANOVA followed by Bonferroni multiple comparisons post-test. Data were approximately normally distributed, and variances were similar between groups. Statistical significance is denoted as * *p* < 0.05, ** *p* < 0.01, *** *p* < 0.001, and **** *p* < 0.0001.

## 3. Results and Discussion

### 3.1. Construction and Characterization of BH-Gel Incorporating Dox Lipo and CXCL12

Doxorubicin (Dox) is the most commonly used first-line treatment for lung cancer [18]. As the commonly utilized drug carrier, liposomes provide several benefits, such as enhanced drug stability and biodegradability. Molecular docking analysis revealed that the nitrogen atom of doxorubicin formed two hydrogen bonds with the O11 and O14 atoms of phosphatidylcholines (PC), with bond lengths of 2.2 Å and 2.1 Å, respectively, highlighting the interaction between Dox and the liposome (Figure 1A). To further confirm the binding between Dox and liposome, the circular dichroism spectroscopy revealed significant conformational changes in PC as the Dox concentration increased (Figure 1B). Fluorescence images further demonstrated that Dox autofluorescence (green) co-localized with the DiI-labeled liposome fluorescence (red), confirming that lecithin successfully encapsulated Dox to form Dox Lipo (Figure 1C). Transmission electron microscopy (TEM) analysis showed that Dox Lipo was homogeneous, spherical, and exhibited a characteristic cup-shaped structure (Figure 1D). And it had a particle size of 141.2 ± 2.7 nm and a PDI of 0.16 ± 0.02, as determined via dynamic light scattering, indicating its potential for efficient Dox delivery to tumor cells (Figure 1E and Appendix A). The zeta potential, which influences particle repulsion and attraction, was measured at an optimal value of −1.394 ± 0.017 mV, which helps to prevent liposome aggregation and fusion (Appendix A). Over 14 days, neither the particle size nor the zeta potential showed significant changes (Figure 1F). Furthermore, Dox leakage from the liposomes remained below 1% over a 7-day period (Figure 1G), confirming the excellent stability of Dox Lipo.

Next, we developed an injectable BH-gel that released the chemoattractant CXCL12 and Dox Lipo to kill tumor cells. PVA is known for its excellent biocompatibility and non-toxic nature, making it suitable for biomedical applications. In our design, the hydrogel backbone was formed through covalent cross-linking between PVA and borax, with CXCL12 and Dox Lipo incorporated to attract and kill tumor cells, respectively (Figure 2A). In aqueous solution, borax dissociates into boronic acid B(OH)_3_ and tetrahedral boronate ions [B(OH)_4_^−^]. The [B(OH)_4_^−^] ions, each featuring four hydroxyl groups, react with the PVA mixture, forming strong interactions with water molecules through boronate ester bonds. The characteristic peak at 1394 cm^−1^ in the FTIR spectrum of the blank gel corresponds to the B─O bond stretching vibration, confirming the formation of boronate ester bonds (Figure 2B). At room temperature, the PVA and borax solutions remained in a uniform flow state. Upon mixing, rapid cross-linking occurred within 5 s, forming boronate ester bonds (Figure 2C). Scanning electron microscopy (SEM) revealed that this hydrogel possessed a three-dimensional porous structure with pore sizes ranging from 300 to 500 nm, which was ideal for encapsulating CXCL12 and Dox Lipo (Figure 2D). Additionally, a dynamic frequency rheological sweep was performed to assess the mechanical strength of BH-gel (Figure 2E). The frequency-dependent profile of the storage modulus (G′) and loss modulus (G″) illustrated the typical behavior of BH-gel. The higher G′ compared to G″ indicated the formation of an elastic gel network. To further investigate the self-healing properties of BH-gel, strain amplitude sweep tests were conducted (Figure 2F). Results showed that the BH-gel maintained a higher G′ than G″ under small strain, indicating a quasi-solid state. As the shearing strain increased to 19%, G′ significantly decreased and fell below G″, signaling a transition to a quasi-liquid state due to the dissociation and collapse of the cross-linked networks. Subsequently, an alternating strain test (10% or 100%) was performed to evaluate the self-healing property of BH-gel, as shown in Figure 2G. When a large strain of 100% was applied, the G′ value dropped below G″, indicating network structure collapse. However, when the strain was reduced to 10%, both G′ and G″ rapidly returned to their original values, with G′ exceeding G″, suggesting successful network reconstruction. Notably, G′ and G″ maintained this trend throughout the test, which was mainly attributed to the dynamic reversible boronate ester bonds in the hydrogel networks [19,20]. Moreover, as the shear rate increased, the viscosity of BH-gel decreased significantly, clearly demonstrating its inherent shear-thinning property (Figure 2H). These results confirmed that BH-gel possessed a classic three-dimensional network structure and exhibited good injectability.

### 3.2. The Injectable BH-Gel Exhibited Excellent Biocompatibility and Promising Electrical Conductivity

Injectability and adherence are important requirements for tumor treatment, as they enable the hydrogel to conform to various cavity shapes and adhere to tissues in vivo, facilitating minimally invasive procedures. When subjected to shear force, BH-gel showed reduced viscosity, allowing for smooth flow through narrow needles. Additionally, it was demonstrated that BH-gel could be continuously and smoothly extruded through a syringe needle to precisely draw the letters “HNU” and fill the mold cavity, underscoring its excellent injectability (Figure 3A). Owing to dynamic cross-linking, the homogeneous BH-gel displayed good shape adaptability, which can be easily molded into various 3D shapes (e.g., heart, crescent, square, and five-pointed star) (Figure 3B and Appendix A). Moreover, when placed between two pieces of pig skin, the BH-gel demonstrated strong adhesion under both dry and wet conditions (Figure 3C). Then, BH-gel was applied to a rabbit’s ear to assess its biosafety. As shown in Figure 3D, there were no signs of edema or erythema on the surface of the rabbit’s ear. Consequently, BH-gel exhibits strong implantation potential and can be utilized in various conditions and sites in vivo, such as subcutaneous areas and cavities. Interestingly, we found that BH-gel exhibited excellent electrical conductivity. When applied to the skin, detection waves from a watch can be transmitted through the gel to the skin to monitor heart rate (Figure 3E). Furthermore, BH-gel can be assembled into a touchscreen pen, allowing for multiple gestures on the smartphone, such as drawing various patterns and making a phone call (Figure 3F and Appendix A). With its favorable electrical conductivity, BH-gel has the potential to serve as a sensor interface and enable dynamic tracking of tumor progression, assess treatment efficacy in situ, and facilitate timely therapeutic adjustments. By integrating these features, BH-gel may significantly enhance personalized cancer care and improve the precision of oncological treatments.

### 3.3. BH-Gel Released CXCL12 to Facilitate Tumor Recruitment

Evidence shows tumor cells tend to migrate towards tissues or organs that are enriched with CXCL12 [21,22]. Thus, we sought to investigate whether BH-gel could attract tumor cells via CXCL12. As seen in Figure 4A, real-time cell migration experiments were conducted to evaluate the movement of A549 cells in response to BH-gel. A549 cells were seeded in the upper chamber, while the BH-gel was placed in the lower chamber. A porous filter that permits the migration of A549 cells separated the upper and lower chambers. The migration profiles of A549 cells were monitored using the RTCA-DP instrument (Figure 4A). In the BH-gel group, the electrical impedance signals continued to rise as more A549 cells migrated downward. In contrast, the blank gel group, which was not loaded with CXCL12, registered only weak electrical signals (Figure 4B). To visually assess migration, the lower chamber was imaged using the EVOS M7000 Imaging System. Fluorescence imaging revealed a significant increase in the number of migrated A549 cells in the BH-gel group, approximately 15-fold compared to the blank gel group (Figure 4C). This suggests that the “bait” CXCL12 released from the BH-gel effectively attracts tumor cells, promoting their active migration toward the gel (Figure 4D). Furthermore, the ability of the BH-gel to capture and engage tumor cells was confirmed in vivo. Blank gel and BH-gel were injected into mice, with tumor cells seeded around the gels, which were then dissected out several days later. As shown in Figure 4E, many tumor cells were captured on the BH-gel, while few were found on the blank gel. In summary, the BH-gel can engage and capture tumor cells through the release of “bait” CXCL12, offering potential for subsequent tumor cell eradication.

### 3.4. Dox Lipo Delivered from BH-Gel Exerted an Optimal Anticancer Effect In Vitro

As tumor cells were recruited, the antitumor effect of the BH-gel was evaluated. To investigate this process, the degradation behavior of BH-gel and the release dynamics of Dox Lipo were initially examined. It was observed that the hydroxyl groups on the PVA surface enhanced the hydrogel’s hydrophilicity, providing it with a strong water absorption capability. This eventually led to the collapse of the hydrogel structure as it swelled [23]. The result presented in Figure 5A revealed that nearly 80% of BH-gel was degraded in PBS within 48 h. This degradation is likely due to the strong hydrophilicity of PVA, which promoted BH-gel to swell and eventually led to the breakage of dynamic boronate ester bonds between PVA and borax. Thus, BH-gel demonstrated excellent degradability, implying its biosafety for in vivo applications. Regarding the release of the Dox Lipo from BH-gel, an initial rapid release was observed during the first 12 h, followed by a slow and sustained release lasting up to 48 h. This confirmed that Dox Lipo could be effectively and fully released from BH-gel, potentially enhancing its therapeutic efficacy. Next, the uptake process of Dox Lipo by A549 cells was investigated. As shown in Figure 5C, flow cytometric analysis displayed a significant difference in fluorescence intensity between the free doxorubicin group and the Dox Lipo group. The cellular uptake rate of free doxorubicin was about 20.45% after 4 h, while Dox Lipo reached up to 75.66% (Figure 5B–D). This result clearly indicated that liposomal encapsulation enhanced the amount of doxorubicin that entered A549 cells. This enhancement is likely due to the structure similarity between the liposome and cell membrane. Both possess amphiphilic phospholipids, facilitating favorable interactions that promote the efficient uptake of liposomes by A549 cells [24]. In agreement with flow cytometric analysis, fluorescence images also verified the enhanced cellular uptake of doxorubicin in the Dox Lipo group (Figure 5E,F). Furthermore, the antitumor effect of Dox Lipo on A549 cells was evaluated. MTT assay showed that free doxorubicin inhibited the growth of A549 cell tumors by only approximately 15% (Figure 5G). In contrast, Dox Lipo significantly suppressed the growth of A549 cells, achieving around 45% inhibition. This represents a 3-fold increase in tumor cell toxicity compared to free doxorubicin, which may be associated with the difference in cellular uptake efficiency. Moreover, the apoptosis assay yielded similar results, with Dox Lipo inducing about a 2.2-fold higher rate of cell death than free doxorubicin (Figure 5H). Specifically, the early apoptosis rate and the late apoptosis increased by 1.8% and 4.01%, respectively, while necrosis surged by 35.4%. Thus, Dox Lipo was fully released from BH-gel within 48 h and rapidly internalized by tumor cells, inducing a strong apoptotic response. Taken together, the above experiments demonstrated that BH-gel successfully lured and captured tumor cells via CXCL12 release, with Dox Lipo subsequently exerting potent tumor-killing effects.

### 3.5. Potent Antitumor and Anti-Metastatic Activity of BH-Gel Against Lung Cancer

As previously discussed, BH-gel showed significant potential to completely obliterate tumors by effectively baiting and killing tumor cells. Building on the promising effects observed in vitro, further investigations were conducted in vivo. To evaluate the tumor cell capture capability and the antitumor efficacy of BH-gel, a metastatic xenograft model of lung cancer was established in BALB/c nude mice through intraperitoneal injection with 5 × 10^6^ A549-GFP cells. Tumor-bearing mice were then intraperitoneally administered BH-gel, Dox Lipo, or saline every 3 days for 3 weeks, resulting in a total of seven doses (Figure 6A). The tumor growth was monitored weekly using an in vivo imaging system, and the animals were sacrificed and dissected on day 28. It was observed that the tumor weights in both the BH-gel group and Dox Lipo group were significantly lower than those in the control group at the experiment’s endpoint, thereby validating the antitumor activity of Dox Lipo in vivo (Figure 6B). Notably, the group receiving BH-gel had significantly smaller tumors compared to the group treated with free Dox Lipo. Due to the lack of effective targeting properties, the insufficient retention of free Dox Lipo at the tumor site impaired its therapeutic effect. In contrast, BH-gel enhanced the retention time and demonstrated potent antitumor activity in vivo, owing to its ability of tumor cell recruitment. Consistently, intravital imaging confirmed these promising results. As presented in Figure 6C,D, the fluorescence signals representing A549-GFP cells were weaker in the Dox Lipo group compared to the control; however, Dox Lipo failed to prevent an increase in fluorescence signals over time. In contrast, the fluorescence signals of A549-GFP tumors in the BH-gel group remained faint and difficult to detect throughout the entire 28 days, indicating excellent tumor eradication efficacy. Additionally, fluorescence signals in both the Dox Lipo group and the control group demonstrated a trend of spreading outward from the initial site, highlighting the risk of tumor metastasis. Then, the metastatic nodules in the mesentery of the mice were counted to assess tumor metastasis. Consistent with the above result, numerous metastatic tumors appeared as white nodules on the mesentery in both the Dox Lipo group (15.8 ± 6.8 nodules/mouse) and the control group (10.5 ± 4.3 nodules/mouse), whereas only a very few nodules were observed in BH-gel group (2.2 ± 2.3 nodules/mouse) (Figure 6E,F), suggesting the robust anti-metastatic capability of BH-gel. Of note, nodule numbers tended to be increased in mice treated with Dox Lipo but did not reach statistical significance. It has been reported that the therapeutic efficacy of chemotherapy varies among different tumor cells due to the ongoing diversification of tumor cell phenotypes during treatment and the pre-existing intratumor heterogeneity [25], which partly accounts for the enhanced tumor invasion associated with Dox Lipo. To address this issue and mitigate the potential risk of tumor metastasis induced by doxorubicin [26], BH-gel was designed to recruit and trap as many tumor cells as possible, thereby reducing the number of metastatic tumor cells. Additionally, BH-gel prolonged the residence time of Dox Lipo and extended its release, providing sufficient time for Dox Lipo to kill the captured tumor cell and exhibited a significantly longer median survival time (Appendix A). As expected, we confirmed that BH-gel significantly enhanced the antitumor efficacy of doxorubicin and effectively suppressed tumor metastasis in vivo, offering a novel promising strategy for the treatment of metastatic lung carcinoma.

### 3.6. In Vivo Assessment of the Systemic Safety Profile of BH-Gel

Finally, the in vivo safety of BH-gel was evaluated. Despite its effective antitumor activity, doxorubicin is known to induce severe dose-dependent cardiotoxicity, manifesting as myofibrillar degeneration and myocytes vacuolization, accompanied by a reduction in body weight [27,28]. Notably, significant decreases in body weight was observed in mice treated with free Dox Lipo during the entire treatment period, suggesting severe systemic toxicity (Figure 7A). In contrast, no noticeable changes in body weight were observed in the other groups, indicating that BH-gel did not induce obvious systemic toxicity. At the endpoint of treatment, major organs, including heart, liver, spleen, lungs, and kidneys, were harvested. H&E-stained cardiac sections revealed prominent tissue injury only in the Dox Lipo group, showing the cardiotoxicity of free Dox Lipo (Figure 7B). Conversely, no abnormalities were detected in the hearts of mice treated with BH-gel, which implied that BH-gel improved the cardiac safety of Dox Lipo. This was primarily due to BH-gel maintaining a high concentration of doxorubicin at the tumor site while reducing the systemic exposure. Additionally, compared with the control group, organ weights were reduced in mice treated with Dox Lipo, whereas no significant organ weight loss was observed in the BH-gel group (Figure 7C–G). This further illustrated the significant systemic toxicity of free Dox Lipo and the biosafety of BH-gel in vivo. Given that liver is the crucial metabolic organ, several serum biochemical indicators were selected to evaluate liver function. Alanine aminotransferase (ALT) and aspartate aminotransferase (AST) are considered markers of liver injury, while alkaline phosphatase (ALP) and total bilirubin (TBIL) represent markers of cholestasis and liver metabolism, respectively [29,30]. It was found that the AST and ALP serum levels were apparently elevated in the Dox Lipo group compared to the control, suggesting that free Dox Lipo may cause liver injury (Figure 7H). However, these levels in the BH-gel group were dramatically decreased to control levels, indicating that BH-gel could alleviate the liver injury induced by Dox Lipo in mice.

Collectively, our research provides a novel biocompatible BH-gel that attenuates doxorubicin-mediated toxicity and exhibits a better biosafety profile in vivo, making it a promising platform for future chemotherapeutic drugs.

## 4. Conclusions

The occurrence of drug-induced invasion and metastasis, coupled with adverse side effects, significantly limits the clinical effectiveness of chemotherapy in lung cancer treatment. Addressing these challenges requires the development of chemotherapy that not only minimizes these harmful effects but also improve patient outcomes. In this study, we developed an injectable bait-and-hook hydrogel (BH-gel) that incorporates doxorubicin liposomes and the tumor chemoattractant CXCL12 for localized drug delivery. The BH-gel, characterized by its shear-thinning properties and advanced electrical conductivity, demonstrated excellent injectability and holds potential for future designs to incorporate dynamic tumor monitoring functionality. Notably, the BH-gel releases a controlled amount of CXCL12, acting as a “bait” to attract circulating tumor cells to the gel. Once recruited, the doxorubicin-loaded liposomes efficiently kill the tumor cells, minimizing systemic toxicity and limiting cancer spread. In vivo experiments demonstrated that locally injected BH-gel exhibited superior antitumor activity against lung tumors compared to doxorubicin liposomes at equivalent doses. This bait-and-hook strategy not only reduces the risk of tumor cell escape but also decreases the adverse side effects associated with doxorubicin, offering a novel and potentially transformative approach for treating metastatic lung cancer.

## Figures and Tables

**Figure 1 pharmaceutics-16-01516-f001:**
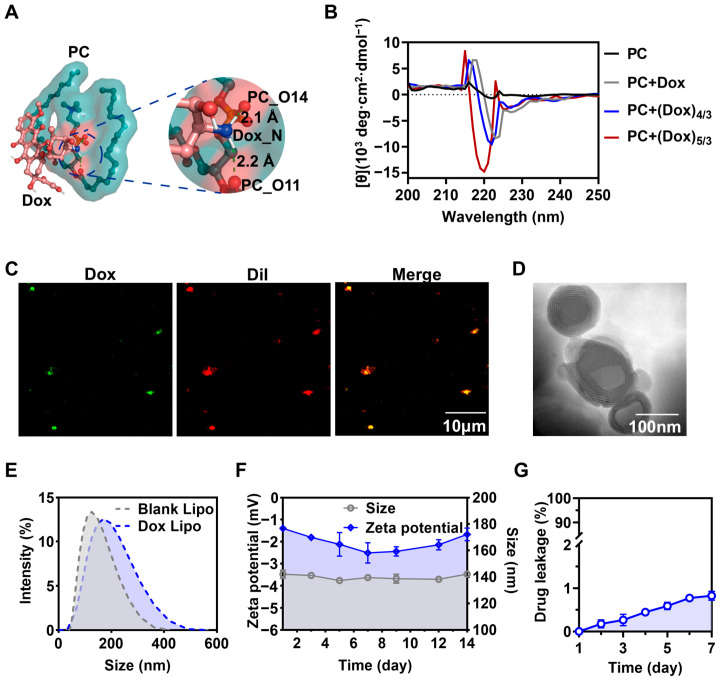
Construction and characterization of Dox Lipo. (**A**) Molecular docking analysis was conducted to study the interaction between Dox and PC. (**B**) The interaction between Dox and PC was confirmed via circular dichroism (CD) spectroscopy. (**C**) Fluorescent analysis of Dox Lipo with a laser scanning confocal microscope (*n =* 3). The Lipo was stained with DiI (red), and Dox produced a fluorescence (green). The merged images were the overlay of two individual images. Scale bar, 10 μm. (**D**) TEM image of Dox Lipo. Scale bars, 100 nm. (**E**) Particle size distribution of Dox Lipo and blank Lipo. (**F**) Changes in particle size and zeta potential of Dox Lipo within 14 days (*n =* 3). (**G**) In vitro Dox leakage from Dox Lipo in cell culture medium at 37 ℃ over 7 days.

**Figure 2 pharmaceutics-16-01516-f002:**
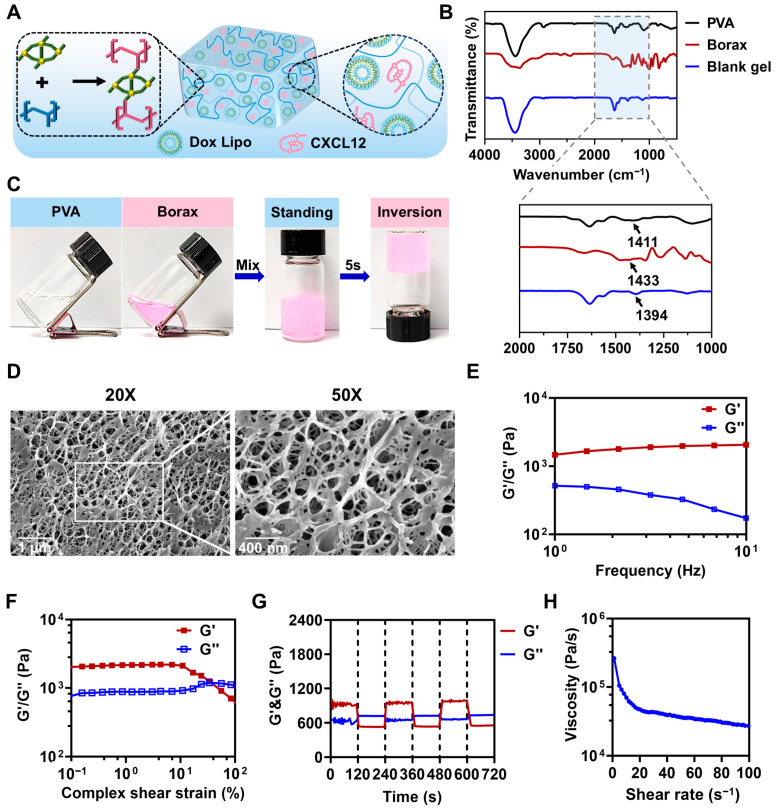
Preparation and characterization of BH-gel based on cross-linking of PVA with borax. (**A**) Schematic illustration of BH-gel cross-linked via boronate ester bonds. (**B**) FT-IR spectra of PVA, borax, and blank gel. (**C**) The gelation of hydrogel occurs after mixing the PVA solution with the borax solution. (**D**) SEM images of blank gel. (**E**) G′ and G″ of BH-gel versus frequency. (**F**) Changes in G′ and G″ with the increasing strain amplitude to 100% at a frequency of 1.0 Hz. (**G**) G′ and G″ under alternating cyclic strain changes between 10% and 100% oscillation strain. (**H**) The viscosity changes in BH-gel at different shear rates (0.1–100 s^−1^).

**Figure 3 pharmaceutics-16-01516-f003:**
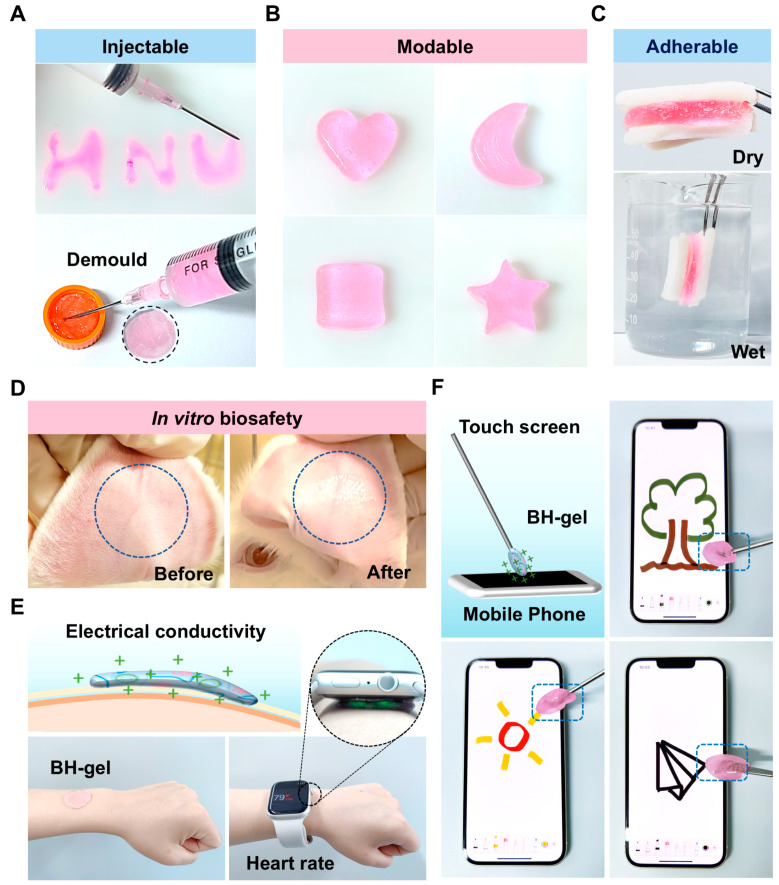
Injectable BH-gel exhibited excellent biocompatibility, deformability, adhesiveness, and electrical conductivity. (**A**) The injectability of BH-gel using a needle syringe was assessed. (**B**) Optical images show the moldable features of BH-gel. (**C**) Photographs showing strong adhesions of the hydrogels with porcine skin in air and under water. (**D**) The biocompatibility of BH-gel. Representative images of the skin appearance before and after gel application to a rabbit’s ear. (**E**,**F**) The electrical conductivity of BH-gel. Demonstration of a BH-gel-based wearable sensor for heartbeat monitoring and a BH-gel-based touchscreen pen used for drawing.

**Figure 4 pharmaceutics-16-01516-f004:**
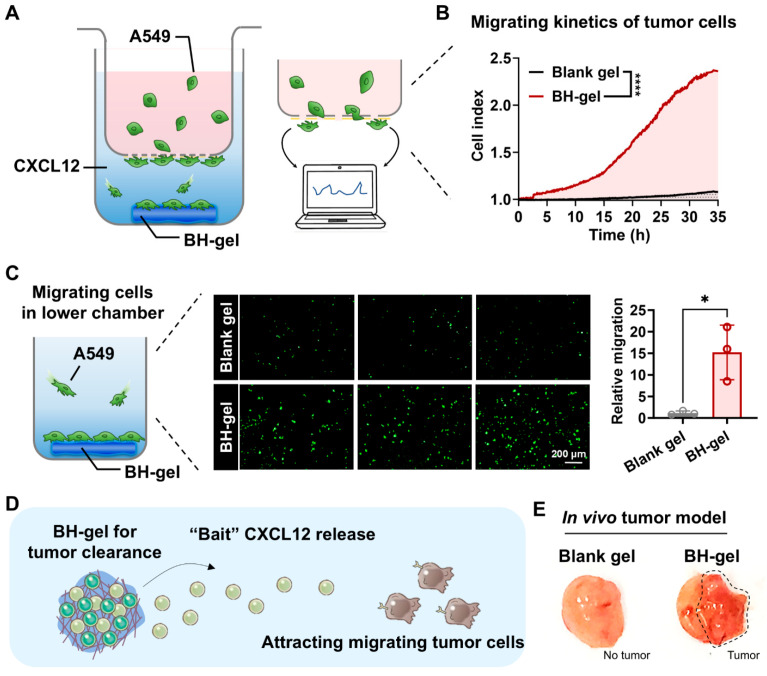
BH-gel releases CXCL12 to facilitate tumor recruitment. (**A**) Schematic illustration of the real-time cell migration assay. BH-gel was placed in the lower chambers, and A549 cells were loaded into the upper chambers. (**B**) Migration kinetics of A549 cells measured via real-time cell migration assays (*n =* 3, **** *p* < 0.0001). (**C**) The cell imaging system demonstrated that BH-gel exhibited a stronger recruitment ability for A549 cells compared to the blank gel, as confirmed with fluorescence quantification (*n =* 3). A549 cells were stained with Calcein-AM. Data are presented as mean ± s.d. Significance levels: * *p* < 0.05 via Student’s *t*-tests. (**D**) Schematic illustration of BH-gel releasing CXCL12 to attract tumor cells. (**E**) Images showed that BH-gel significantly attracted tumor cells in vivo. Blank gel and BH-gel were injected subcutaneously into the area adjacent to the tumor, and both gels were removed and photographed after 2 days.

**Figure 5 pharmaceutics-16-01516-f005:**
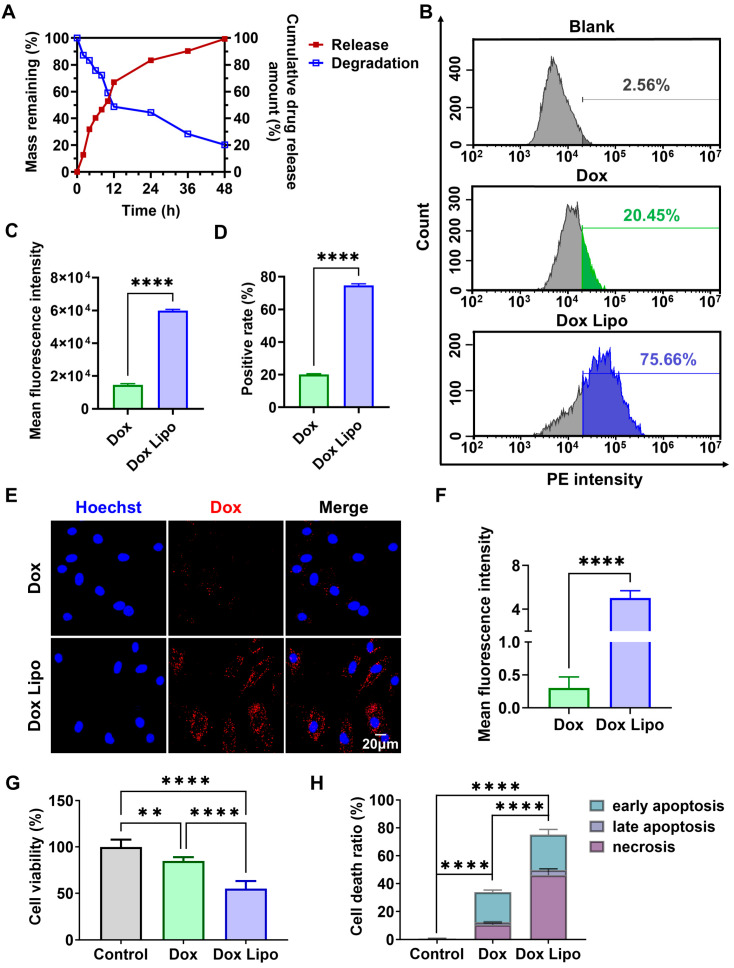
Dox Lipo delivered from BH-gel exerted an optimal anticancer effect. (**A**) Quantification of BH-gel degradation dynamics and Dox Lipo release rate. (**B**) Flow cytometry plots showing cellular uptake of Dox Lipo by A549 cells after 4 h of incubation. (**C**,**D**) Fluorescence intensities and percentages of positive cells were determined via flowcytometry at the indicated times (*n* = 3). (**E**) CLSM images showing the cellular uptake of Dox Lipo (32 µM) by A549 cells after 4 h. The nuclei were stained with Hoechst, and Dox Lipo produced red fluorescence. The merged images represent the overlay of two separate images. Scale bar, 20 µm. (**F**) Statistical analyses of relative Dox fluorescent intensities in A549 cells were based on ImageJ quantification of randomly selected cells (*n* = 3) from the corresponding fluorescence images. (**G**) Cell viability of A549 cells after exposure to Dox and Dox Lipo at a concentration of 32 μM for 24 h. (**H**) Annexin V-FITC/PI assay for apoptosis detection of A549 cells under the treatment of Dox and Dox Lipo for 24 h. The data are shown as mean ± s.d., ** *p* < 0.01, and **** *p* < 0.0001 via Student’s *t*-tests or one-way ANOVA test.

**Figure 6 pharmaceutics-16-01516-f006:**
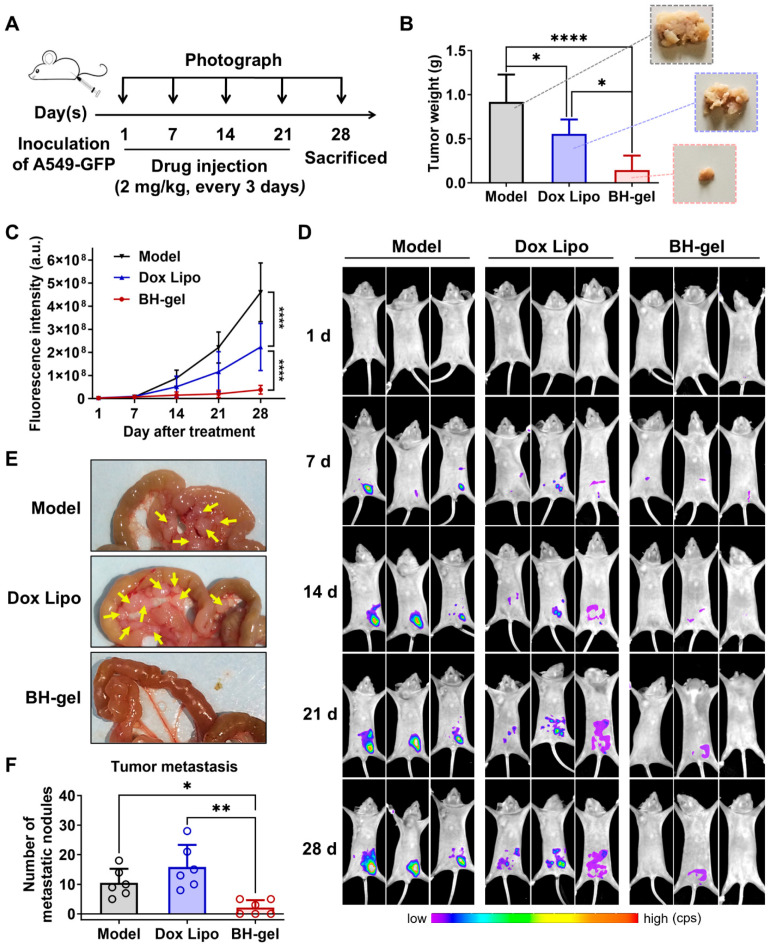
In vivo therapeutic efficacy of BH-gel against lung tumors. (**A**) Experimental procedure of tumor induction and therapeutic regimen. (**B**) Final tumor weights and photographs of metastatic nodules collected on several peritoneal organs after different treatments for 28 days (*n* = 6 per group). (**C**) Quantitative fluorescence intensity of GFP in tumor-bearing mice with or without any treatments for 4 weeks. (**D**) The images of GFP-expressing tumor-bearing mice during four-week-treatments of various formulations. At baseline (1 day), all groups showed equal abdomen fluorescence indicative of equal tumor burden. By week 1 to 4, the tumor burden was reduced in the mice treated with BH-gel compared with controls. (**E**) The location of metastatic nodules on gastrointestinal tracts. Arrows indicate metastatic regions. (**F**) The number of metastatic nodules was counted in control and Dox-treated mice (*n* = 6 per group). The data are shown as mean ± s.d.; * *p* < 0.05, ** *p* < 0.01, and **** *p* < 0.0001 via one-way ANOVA test or two-way ANOVA test.

**Figure 7 pharmaceutics-16-01516-f007:**
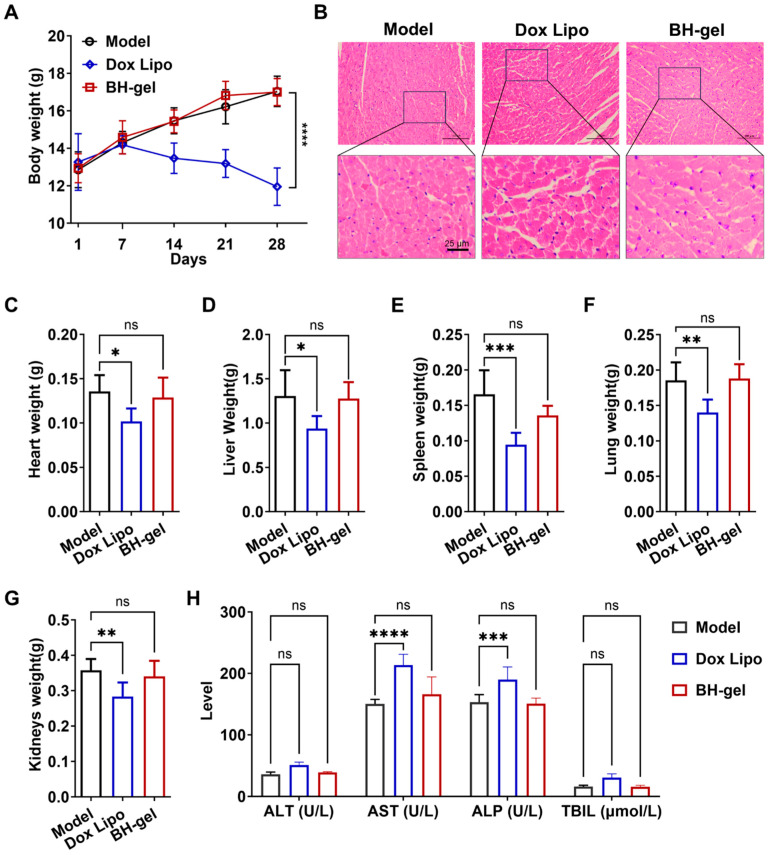
BH-gel shows no significant toxicity or side effects. (**A**) Body weights recorded over 28 days of different treatments (*n* = 6 per group). (**B**) H&E-stained heart tissues from control and treated tumor-bearing mice on day 28. (**C**–**G**) Weights of the heart, liver, spleen, lungs, and kidneys after 28 days of treatment (*n* = 6 per group). (**H**) Serum biochemical parameters related to liver function were analyzed (*n* = 6; ALT, AST, ALP, and TBIL). Data are presented as mean ± s.d. Significance is indicated as follows: * *p* < 0.05, ** *p* < 0.01, *** *p* < 0.001, **** *p* < 0.0001, and ns *p* > 0.05 via one-way ANOVA test.

## Data Availability

All data are available in this article. They can be requested from the author for correspondence.

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
