# Peer review of "A Bait-and-Hook Hydrogel for Net Tumor Cells to Enhance Chemotherapy and Mitigate Metastatic Dissemination"

_pharmaceutics, 2024, doi:10.3390/pharmaceutics16121516_

Round 1

Reviewer 1 Report

Comments and Suggestions for Authors
  • The current paper presents an interesting design of a hydrogel for the co-delivery of an anti-metastatic agent and doxorubicin-loaded liposomes in the context of lung cancer treatment. The following clarifications are required:
  • The liposome composition is not among the standard formulations employed in this type of systems. Please justify 
  • Similarly, the authors should provide a sound justification for the selection of PVA-based hydrogels
  • In the physicochemical characterization, the low values of zeta potential obtained would not be enough to prevent aggregation. Please discuss this attending to the polydispersity measured by dynamic light scattering 
  • The authors propose the formation of boronate ester bonds as the mechanism for hydrogel crosslinking. Please reference this as well as the hypothesis that the formation of these bonds is compatible with the self-healing behaviour observed in the rheological characterization. Please use consistent nomenclature (boric vs borate vs boronate bonds)
  • Explain how the very rapid crosslinking (5 seconds) impacts injectability
  • I don't agree with the claim of biosensing capabilities for the hydrogel; the results provided don't seem to evidence any analytical or transducing properties. Please revise
  • Details of the experiments with rabbit tissue should be included
  • The authors claim an almost full degradation of the hydrogel within 2 days in PBS. Please explain how impactful will be the presence of such short-term hydrogel: 1) in effectively capturing a significant mass of tumour cells as compared to the release of free CXCL12, and 2) as compared to a treatment with loaded liposomes and CXCL12 directly
Comments on the Quality of English Language

The manuscript reads well and would probably require minor editing

Reviewer 2 Report

Comments and Suggestions for Authors

The manuscript entitled “A bait-and-hook hydrogel net tumor cells to enhance chemo therapy and mitigate metastatic dissemination” is interesting, innovative, and well-structured. The authors incorporated doxorubicin liposomes and CXCL12 into BH-gel, which exhibited significant antitumor activity against lung tumors and effectively mitigated metastatic dissemination. However, several aspects of the manuscript could benefit from further refinement.

Comments:

1.     Provide a Graphical abstract.

2.     Could the authors clarify whether the encapsulation efficiency of the liposome was tested when loading Dox? What is the ratio of Dox to liposomes when preparing Dox Lipo? Additionally, the details regarding the liposome are missing. Please provide the source of the liposome along with relevant information.

3.     Please include liposome-based drug encapsulation for effective drug delivery in the Introduction section. Refer to and cite the following article: “Cui L, Renzi S, Quagliarini E, et al. Efficient Delivery of DNA Using Lipid Nanoparticles. Pharmaceutics. 2022;14(8):1698.”.

4.     The authors reported that the Dox liposomes released the drug from BH-gel in a sustained manner, lasting up to 48 hours. Given that the release kinetics of drugs can substantially influence treatment efficacy, especially when it comes to cancer treatment, prolonged release may offer additional advantages for effective tumor control, I would like to ask whether the authors evaluated release over longer duration (more than 48 hours)? And what percentage of the total drug was released within the first 48 hours?

5.     In the cellular uptake experiment (Figure 5.E), the uptake efficiency of Dox by A549 cells appears to be quite low. Could the authors specify the concentration of Dox used in this experiment? The manuscript only mentions the concentration of Dox liposomes, which is 32 μM. Additionally, could the authors elaborate on the rationale behind choosing this specific concentration of 32 μM for Dox lipo? Doxorubicin is known for its ability to localize in the nucleus due to its mechanism of action, primarily involving DNA intercalation and inhibition of topoisomerase II. However, in this study, it appears that Dox does not localize in the nucleus when cells are incubated with the Dox alone. Can the authors provide an explanation for this unexpected observation?

Round 2

Reviewer 1 Report

Comments and Suggestions for Authors

Following the revision by the authors, I believe this manuscript is now ready for publication.